# The Effects of Foam Rolling Training on Performance Parameters: A Systematic Review and Meta-Analysis including Controlled and Randomized Controlled Trials

**DOI:** 10.3390/ijerph191811638

**Published:** 2022-09-15

**Authors:** Andreas Konrad, Masatoshi Nakamura, David George Behm

**Affiliations:** 1Institute of Human Movement Science, Sport and Health, Graz University, 8010 Graz, Austria; 2School of Human Kinetics and Recreation, Memorial University of Newfoundland, St. John’s, NL A1C 5S7, Canada; 3Faculty of Rehabilitation Sciences, Nishi Kyushu University, 4490-9 Ozaki, Kanzaki 842-8585, Saga, Japan

**Keywords:** roller-massage, strength, myofascial technique, jump, stretch tolerance

## Abstract

Foam rolling (FR) is a new and popular technique for increasing range of motion. While there are a few studies that demonstrate increased performance measures after an acute bout of FR, the overall evidence indicates trivial performance benefits. As there have been no meta-analyses on the effects of chronic FR on performance, the objective of this systematic meta-analytical review was to quantify the effects of FR training on performance. We searched PubMed, Scopus, the Cochrane library, and Web of Science for FR training studies with a duration greater than two weeks, and found eight relevant studies. We used a random effect meta-analysis that employed a mixed-effect model to identify subgroup analyses. GRADE analysis was used to gauge the quality of the evidence obtained from this meta-analysis. Egger’s regression intercept test (intercept 1.79; *p* = 0.62) and an average PEDro score of 6.25 (±0.89) indicated no or low risk of reporting bias, respectively. GRADE analysis indicated that we can be moderately confident in the effect estimates. The meta-analysis found no significant difference between FR and control conditions (ES = −0.294; *p* = 0.281; I^2^ = 73.68). Analyses of the moderating variables showed no significant differences between randomized control vs. controlled trials (Q = 0.183; *p* = 0.67) and no relationship between ages (R^2^ = 0.10; *p* = 0.37), weeks of intervention (R^2^ = 0.17; *p* = 0.35), and total load of FR (R^2^ = 0.24; *p* = 0.11). In conclusion, there were no significant performance changes with FR training and no specific circumstances leading to performance changes following FR training exceeding two weeks.

## 1. Introduction

Foam rolling (FR) and stretching are regularly used during warm-up routines in sport settings to prepare the body for the following performance task. A single foam rolling (FR) exercise can increase the range of motion (ROM) of a joint immediately after the treatment [1,2] but also for durations of up to 30 min post-treatment [3]. Concerning a single FR treatment and its effects on sports performance (e.g., strength, jump height), a recent meta-analysis [4] reported negligible effects in jump and strength performance, but a tendency for immediate improvements (*p* = 0.06) in sprint performance (+0.7%; ES = 0.28). Hence, according to this meta-analysis [4], as well as other reviews [5], a single FR exercise conducted as a warm-up likely has no significant impact on sports performance.

When FR is compared to stretching, meta-analyses show that both treatments are similarly effective in acutely increasing ROM [6]. However, FR showed more favorable effects compared to stretching on performance parameters, at least under specific circumstances, such as when compared to static stretching, when applied to some selected muscles (i.e., quadriceps, triceps surae) or strength tasks (e.g., maximum voluntary contraction torque), when applied for longer than 60 s, or when the FR included vibration [7]. Hence, the authors of theses reviews [6,7] concluded that FR might the better choice as a warm-up, compared to stretching alone (i.e., if performed without a comprehensive warm-up [8,9]).

Regarding long-term (i.e., chronic/training) stretch training and its various methods (e.g., static stretching), chronic increases in ROM are regularly reported [10,11]. With regard to performance parameters, some studies report no changes following long-term stretch training (e.g., [12,13]), whilst others report an increase in performance (e.g., [14,15]). However, when considering the long-term effects of FR, much less is known. One recent review reported conflicting results for FR training and its effect on ROM [16]. However, a further review [17] on this topic performed a meta-analysis and showed that long-term FR interventions can increase joint ROM in young, healthy participants with a moderate effect size (ES = 0.823). However, this increase may be muscle and/or joint dependent, as FR on the triceps surae muscle did not increase ankle dorsiflexion ROM. Additionally, an intervention duration of more than four weeks was needed to observe significant changes in ROM [17]. However, to date, little evidence is available on the effects of FR training on ROM; Konrad et al. [17] and Pagaduan et al. [16] reported eleven and eight eligible studies on that topic, respectively.

Additionally, Pagaduan et al. [16] analyzed the training effects of FR on performance parameters and concluded, based on six eligible studies, that FR has neither detrimental nor enhancing effects on performance. However, these authors did not perform a meta-analysis, which would definitely give more quantitative details as to whether FR can have an impact on performance parameters.

Hence, this systematic review and meta-analysis aims to overcome this limitation, and examines whether FR training interventions can change performance parameters in healthy participants by considering moderating variables such as age, weeks of intervention, total load of FR intervention, and study design. We hypothesized that FR training does not alter performance parameters, even with consideration of the moderating variables.

## 2. Materials and Methods

This review was conducted according to the PRISMA guidelines and the suggestions from Moher et al. [18] for systematic reviews with a meta-analysis.

### 2.1. Search Strategy

An electronic literature search was performed of PubMed, Scopus, Web of Science, and the Cochrane library. Papers were considered if they were published before 29 July 2022. The terms used to detect long-term foam rolling intervention studies were similar to those used in a recent review on the long-term effects of stretching on ROM (i.e., “chronic effects”, “training effects”, “effects”, “long-term”, and “intervention”) [19]. Moreover, to identify studies dealing with foam rolling, the search terms “foam rolling”, “self-myofascial release”, “roller massage”, and “foam roller” were used, according to previous meta-analyses [7,20]. To detect studies with performance parameters, we used a search code similar to that used in a previous study [21]. The search code for all four databases was (“chronic effects” OR “training effects” OR effects OR “long-term” OR intervention) AND (“foam rolling” OR “self-myofascial release” OR “roller massage” OR “foam roller”) AND (performance OR strength OR force OR hypertrophy OR power OR torque OR height OR RFD OR “rate of force development” OR “jump” OR maximum OR maximal) (please see also Appendix A). The systematic search was conducted by two independent researchers (AK, MN). In the first step, all the hits were screened according to their title and abstract. If the content of a study remained unclear, the full text was screened to identify the relevant papers. Following this independent screening process, the researchers compared their findings. Disagreements were resolved by jointly reassessing the studies against the eligibility criteria.

### 2.2. Inclusion and Exclusion Criteria

This review considered studies that investigated the long-term training effects (>2 weeks [19]) of foam rolling on performance parameters such as strength, power, and jump performance in healthy participants. We included peer-reviewed original studies in English and German. Moreover, studies were included when they were either randomized controlled trials or controlled trials. This means that we excluded studies which dealt with the acute effects of foam rolling (or interventions shorter than <2 weeks), investigated any combined treatment (e.g., foam roller combined with stretching), or had another treatment as a control condition (e.g., stretching). Moreover, we excluded review papers, case reports, special communications, letters to the editor, invited commentaries, conference papers, and theses.

### 2.3. Extraction of the Data

From the included papers, we extracted the characteristics of the participants, the sample size, the study design, the characteristics of the intervention (i.e., weeks of intervention, frequency of intervention per week, duration of each training session per muscle tendon unit, pressure of the foam roller, frequency with which the foam roller was applied), and the results of the main variables (performance parameters). For the performance parameters, pre- and post-intervention values plus standard deviations of the foam rolling group (and, if applicable, of the control group) were extracted. If some of the required data were missing from the included studies, the authors of the studies were contacted via email or similar channels (e.g., Research Gate).

### 2.4. Statistics and Data Synthesis

The meta-analysis was performed using Comprehensive Meta-Analysis software (Version 3; USA), according to the recommendations of Borenstein et al. [22]. By applying a random-effect meta-analysis, we assessed the effect size in terms of the standardized mean difference. If any study reported more than one effect size, the mean of all the outcomes (effect sizes) within one study was used for the analysis and was defined as combined (as suggested by Borenstein et al. [22]). Moreover, by applying a mixed-effect model, we performed subgroup analyses. Although there is no general rule of thumb [22], we only performed subgroup analyses when there were ≥3 studies included in the respective subgroups. Consequently, we were unable to perform subgroup analyses on activity level (highly active vs. recreational), sex, or muscle used during the performance tests. However, a subgroup analysis for the study design (RCT vs. CT) was performed. To determine differences between the effect sizes of the subgroups, Q-statistics were applied [22]. Moreover, to assess possible relations in the moderating variables we conducted a meta-regression (i.e., age of the participants, weeks of intervention, total load of FR intervention). According to the recommendations of Hopkins et al. [23], the effects for a standardized mean difference of <0.2, 0.2–0.6, 0.6–1.2, 1.2–2.0, 2.0–4.0, and >4.0 were defined as trivial, small, moderate, large, very large, and extremely large, respectively. I^2^ statistics were calculated to assess the heterogeneity among the included studies, and thresholds of 25%, 50%, and 75% were defined as having a low, moderate, and high level of heterogeneity, respectively [24,25]. An alpha level of 0.05 was defined for the statistical significance of all the tests.

### 2.5. Risk of Bias Assessment and Methodological Quality

The methodological quality of the included studies was assessed using the PEDro scale. In total, 11 methodological criteria were rated by two independent researchers (A.K., M.N.) and were assigned either one point or no points. Hence, higher scores indicated that the study had a higher methodological quality. In the case of a conflict between the two researchers, the methodological criteria were reassessed and discussed. Moreover, statistics of the Egger’s regression intercept test and visual inspection of the funnel plot were applied to detect possible publication bias.

### 2.6. Confidence in the Cumulative Evidence

Grading of Recommendations, Assessment, Development and Evaluations (GRADE) rating analysis was used to assess the quality of the outcomes by using the GRADEpro Guideline Development Tool software (gradepro.org; McMaster University, Hamilton, Canada). GRADE assesses four general levels of evidence quality: very low, low, moderate, and high. For GRADE analysis, six evaluation components were adopted (study design, risk of bias, inconsistency of results, indirectness, imprecision, and others).

## 3. Results

### 3.1. Results of the Search

Overall, after removal of duplicates, 382 papers were screened, from which seven papers were found to be eligible for this review [26,27,28,29,30,31,32]. However, following the additional search of the references (identified through the reference lists) and citations (identified through Google Scholar) of the seven already included papers, one more paper [33] was identified as relevant. Therefore, in total, eight papers were included in this systematic review and meta-analysis. The search process is illustrated in Figure 1.

Overall, 25 effect sizes could be extracted from the eight eligible studies. In summary, 245 participants with a mean age of 22.9 (±4.2 years) participated in the included studies. Table 1 presents the characteristics and Table 2 the outcomes of the eight studies.

### 3.2. Assessment of Risk of Bias and Methodological Quality

Figure 2 shows a funnel plot, including all eight studies in this meta-analysis. A visual inspection of the funnel plot and the Egger’s regression intercept test (intercept 1.79; *p* = 0.62) indicates no reporting bias. The methodological quality, as assessed with the PEDro scale, revealed a range of scores between 5 and 8 points (out of 10) for all the included studies. The average PEDro score was 6.25 (±0.89), indicating a low risk of bias [34,35] (Table 3). The two assessors agreed on 90.9% out of the 88 criteria (8 studies × 11 scores). The mismatched outcomes were discussed, and the assessors agreed on the scores presented in Table 3.

### 3.3. Overall Effects

The meta-analysis of performance parameters showed that there was no significant difference between the effects of an FR intervention and the control conditions (ES = −0.294; Z = −1.079; CI (95%) −0.828 to 0.240; *p* = 0.281; I^2^ = 73.68). Figure 3 presents the forest plot of the meta-analysis, sorted according to the standard difference in means beginning with the lowest value (−1.468) up to the highest value (1.340).

### 3.4. Moderating Variables

The subgroup analysis showed no significant difference between the study designs (RCT vs. CT) according to the Q-Statistics (Q = 0.183; df = 1; *p* = 0.67) (see Figure 4). Furthermore, the meta-regression showed no relationship between the effect sizes to age (R^2^ = 0.10; *p* = 0.37), weeks of intervention (R^2^ = 0.17; *p* = 0.35), and total load of FR intervention (R^2^ = 0.24; *p* = 0.11).

### 3.5. Confidence in Cumulative Evidence

In terms of study design, we included controlled and randomized controlled trials for the GRADE analysis. Risk of bias, indirectness, and imprecisions showed no serious shortcomings. However, we identified serious inconsistency, as two studies reported the significant improvement of performance parameters following FR training, one study showed an impairment, and the remaining study showed no change. As a consequence, the analysis shows that we can be moderately confident in the effect estimates. This implies that the true effect is likely to be close to the estimate of the effect.

## 4. Discussion

The purpose of this meta-analysis was to investigate whether FR training over several weeks can change performance parameters (e.g., maximum voluntary contraction torque, jump height). Moreover, we tested whether moderating variables such as age, weeks of intervention, total load of FR intervention, and study design have an impact on the study outcomes. Neither the main meta-analysis of the eight eligible studies (25 effect sizes in total), which showed a low risk of bias [35,36], nor the analysis of the moderating variables (i.e., subgroup analysis, meta-regression) showed any significant change in performance parameters in the FR intervention groups compared to the control groups.

Concerning other chronic flexibility treatments, such as static stretch training, previous studies report conflicting results on performance parameters. While some report no significant changes [12,13,36,37,38], others report an increase in performance parameters [15,39,40,41,42,43]. However, in most of the studies which reported an increase in performance, either an inactive population was used as a sample [39], or a high volume or high intensity stretching protocol was applied [14,15]. Concerning the inactive population, simply holding the position during a stretching exercise with the contralateral limb (e.g., when considering the classical quadriceps stretch in a standing position), rather than the stretching stimulus, can result in such an increase in performance in the long-term [39]. However, Panidi et al. [15] recently found an increase in jump performance in adolescent female volleyball players. However, these athletes performed a very comprehensive stretching protocol (>45 min per week of only the calf muscles), which ultimately increased the performance variables. Consequently, it can be assumed that time under tension is a crucial variable in stretching for performance parameters. The studies included in this meta-analysis did not perform a very comprehensive FR protocol. FR durations within the included studies ranged from 1080 s (=18 min) [33] to 3685 s (=61 min) [32]. This is not highly related to the aforementioned high-volume stretching study, which showed increases in performance. Early stretching studies using animal models reported significant muscle anabolic effects with sustained intermittent stretching, although the durations were far longer than could be tolerated by humans [43]. Greater time under tension can be a potent stimulus for strength and hypertrophy gains with resistance training [44]. Thus, future studies on FR should apply a high-volume stimulus to investigate whether performance parameters can be affected by the comprehensive time under tension. This can be achieved by increasing the duration of the FR bouts, increasing the number of FR training sessions per week, or by even extending the FR duration for more than the maximum observed period (i.e., 8 weeks; see Table 1). However, meta-regression analyses have not demonstrated a significant relationship between the effect sizes to the weeks of intervention (R^2^ = 0.17) or to the total load of FR intervention (R^2^ = 0.24). This is likely due to the low number of eligible studies.

Concerning the other moderating variables, we did not find a significant relationship between the effects sizes to age (R^2^ = 0.10). The likely reason that there was no significant relationship is that the age range (i.e., of the mean ages) of the sample of eligible studies was quite narrow, ranging from 15 to 29.8 years. Consequently, future studies should conduct FR training studies with an elderly population as well.

If FR is applied to the hamstrings, quadriceps, or calf muscles, a position related to a plank position must be taken. A previous study showed that the muscle activity (i.e., electromyography) of the core muscles with a plank or reverse plank position is similar to that with quadriceps and hamstrings FR, respectively [45]. Thus, it can be assumed that the participants in the eligible studies of this meta-analysis might have improved their performance, but in their core muscles rather than in the tested/treated muscle. The studies that exclusively rolled the quadriceps, hamstrings, or calf muscles did not test for core strength (see Table 2). Meanwhile, Junker and Stöggl [29], who applied the FR exercises to all the major lower body parts (i.e., quadriceps, hamstrings, calves, IT band, glutes) in fact showed a significant increase in the lateral Bourbon trunk muscle strength test; however, this change did not differ from that seen in the control group or the core strength group. Considering the ventral or dorsal side of the Bourbon trunk muscle strength test, no changes were reported after the eight weeks of FR intervention [29]. However, it can again be speculated that more time under tension would have led to significant improvements, at least in core performance.

Moreover, it is important to investigate which mechanisms are involved in long-term FR training, which, for example, can induce an increase in ROM, as shown in a recent meta-analysis [17]. To date, to the best knowledge of the authors, there are only two studies which assess neurological (tolerance to stretch) but also structural (muscle stiffness) parameters to identify possible mechanisms [30,46]. Foam rolling is a type of soft tissue self-massaging that aims to release the soft tissue from the traction exerted by a fascia that has become either inelastic or adherent to adjacent tissues due to injury or pathology [47,48]. Although it is not clear whether foam rolling releases myofascia [49], acute increases in soft tissue elasticity and pain thresholds, and subsequently stretch tolerance, have also been observed [2], and it is assumed that altered pain perception (rather than changes in muscle stiffness) is also a possible mechanism for long-term increases in ROM [46]. Although a decrease in muscle stiffness has the potential to decrease injury prevalence [50], such a decrease might also accompany a decrease in force production, at least following an acute bout of stretching [51].

This systematic review and meta-analysis have some limitations. Firstly, different study designs, such as CT and RCT, were considered in this meta-analysis. However, no significant differences between these study designs were detected in our subgroup analysis (see Figure 4); as such, we are confident that this did not affect our results to a great extent. Secondly, the intervention durations differed between the included studies, ranging from four to eight weeks. However, our meta-regression analyses showed no significant relationship between the intervention duration and the effect sizes. Thirdly, the performance parameters considered in this meta-analysis were similar but not the same, which might have affected our results. Fourthly, our analyses considered a spectrum of healthy populations (e.g., untrained, trained); as such, no general conclusion for the general population can be given. Some of the variables mentioned in this limitations section might explain the moderate to high heterogeneity of our main meta-analysis (I^2^ = 73.68).

## 5. Conclusions

In conclusion, our meta-analysis showed no significant changes in performance when FR training is applied for several weeks. Additionally, according to our evidence (i.e., meta-regression; subgroup analyses), no specific circumstances lead to a change in performance following a bout of FR training. Future studies should apply high-volume FR interventions and investigate the potential increase in core strength due to higher levels of muscle activation, especially during quadriceps and hamstring FR.

## Figures and Tables

**Figure 1 ijerph-19-11638-f001:**
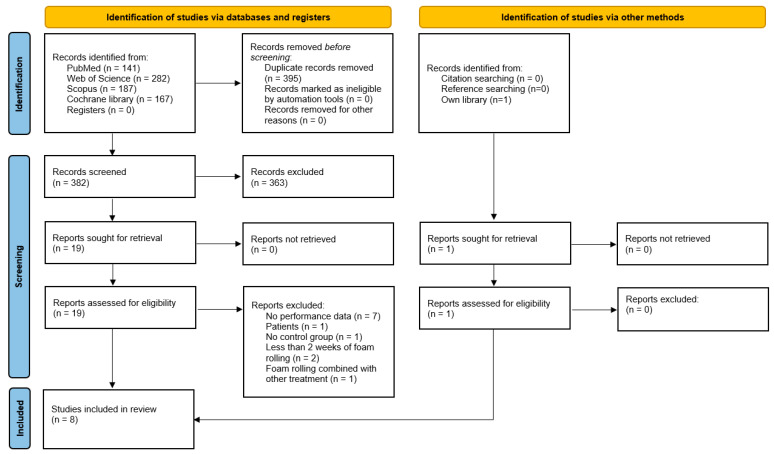
Prisma flowchart.

**Figure 2 ijerph-19-11638-f002:**
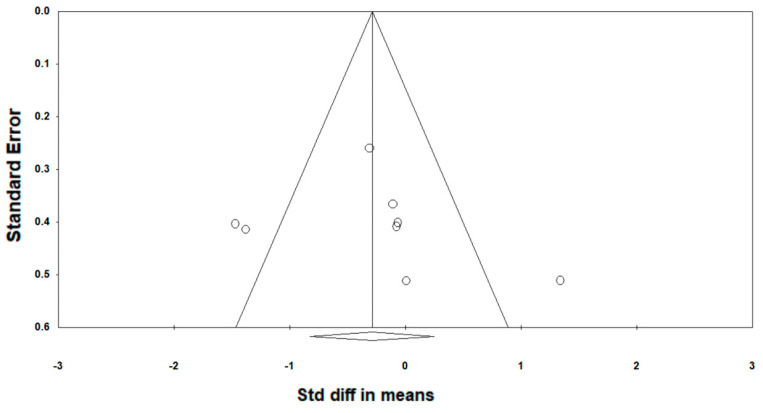
Funnel plot analysis. Circles represent the data from eight individual studies.

**Figure 3 ijerph-19-11638-f003:**
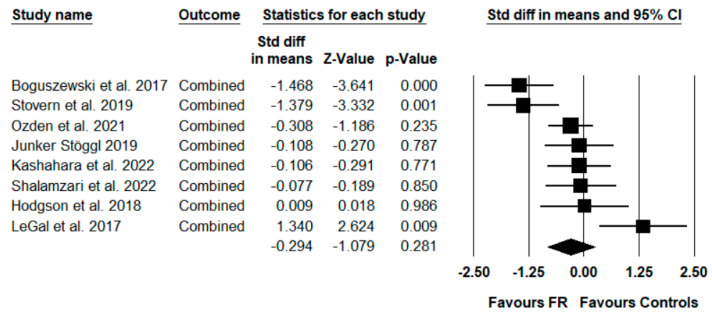
Forest plot presenting the eight included studies (squares) that investigated the effects of foam rolling (FR) on performance parameters (ROM) [26,27,28,29,30,31,32,33]. Std diff in means = standardized difference in means; CI = confidence interval; combined = mean of the selected outcomes of one study. Diamond represents mean standard difference in means of the eight studies.

**Figure 4 ijerph-19-11638-f004:**
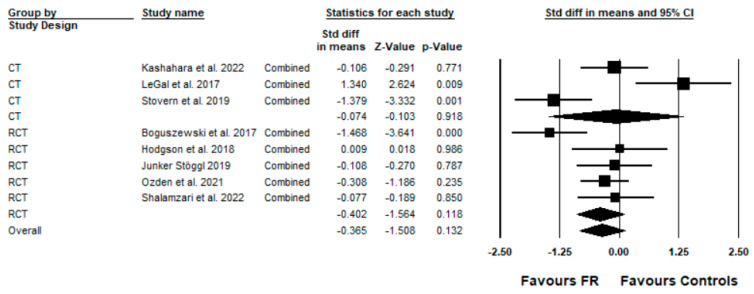
Forest plot presenting the subgroup analysis of study design (controlled trials (CT) vs. randomized controlled trials (RCT)) from the eight included studies (squares) that investigated the effects of foam rolling (FR) on performance parameters (ROM) [26,27,28,29,30,31,32,33]. Std diff in means = standardized difference in means; CI = confidence interval; combined = mean of the selected outcomes of one study. Diamond represents mean standard difference in means of the studies.

**Table 1 ijerph-19-11638-t001:** Characteristics of the included studies (n = 8). IG = intervention group; CG = control group; nr = not reported; VAS = visual analogue scale; ROM = range of motion; bl = bilateral; RCT = randomized controlled trial; CT = controlled trial.

Study	Participants	Intervention Duration (Weeks)	Training Sessions per Week	Duration (s) per Training Session and Muscle Group	Total Load over the Entire Training Program (s)/Muscle Group	Frequency of Foam Rolling (One Direction) (s)	Pressure of Roller	RCT or CT
Boguszewski et al. [26]	N = 37 physically active women; IG: n = 19 (age = 22.8 ± 2.3), CG: n = 18 (age 24.4 ± 1.6)	8	2	nr	nr	nr	nr	RCT
Hodgson et al. [27]	N = 23 recreational actives; 13 males (25.1 ± 2.9); 10 females (age 24.9 ± 4.3)	4	3 or 6	120	1440 or 2880	1	7/10 VAS	RCT
Junker and Stöggl [29]	N = 26 recreational active male and female; IG: n = 14 (age 30.5 ± 10.2), CG: n = 12 (age 29.1 ± 6.9)	8	2	95	1520	nr	Mild to moderate pain (7/10 VAS)	RCT
Kasahara et al. [30]	N = 30 male university students (age 21.6 ± 2.4)	6	2	180	2160	1	As much body weight as possible	CT
Le Gal et al. [28]	N = 11 advanced level male and female tennis players (age 15 ± 3)	5	3	180	2700	nr	Under the threshold of pain	CT
Ozden et al. [31]	N = 60 healthy male and female; IG: n = 30 (age 21.7), CG: n = 30 (age 23.97)	4	3	120	1440	3	As much body weight as possible	RCT
Shalamzari et al. [32]	N = 24 male college athletes IG: n = 20 (age 24.8 ± 2.1), CG: n = (age 25.1 ± 1.9)	8	3	Progressive 45 s to 240 s	3685	nr	nr	RCT
Stovern et al. [33]	N = 34 recreationally active male and female; IG: n = 20 (age 20.8 ± 1.70), CG: n = (age 20.8 ± 1.19)	6	3	60	1080	nr	nr	CT

**Table 2 ijerph-19-11638-t002:** Outcomes of the included studies (n = 8). IG = intervention group; CG = control group; nr = not reported; VAS = visual analogue scale; ROM = range of motion; bl = bilateral; RCT = randomized controlled trial; CT = controlled trial.

Study	Outcome
Boguszewski et al. [26]	Core muscle strength and stability test; functional movement screen
Hodgson et al. [27]	vertical jump height; Maximum voluntary contraction of hamstrings; maximum voluntary contraction of quadriceps
Junker and Stöggl [29]	Bourban trunk muscle strength test; standing long jump; single leg triple hop for distance
Kasahara et al. [30]	Maximum voluntary contraction of plantar flexors
Le Gal et al. [28]	Tennis serve accuracy; tennis serve velocity
Ozden et al. [31]	Biceps brachii muscle strength; modified pull-up test; closed kinetic chain upper extremity stability test; push-up test
Shalamzari et al. [32]	Hamstrings-to-quadriceps strength ratio
Stovern et al. [33]	*t*-test; vertical jump height

**Table 3 ijerph-19-11638-t003:** PEDro scale of the included studies; ^a^ = was not counted for the final score; 1 = one point awarded; 0 = no point awarded. 1. Eligibility criteria were specified. 2. Subjects were randomly allocated to groups (in a crossover study, subjects were randomly allocated an order in which treatments were received). 3. Allocation was concealed. 4. The groups were similar at baseline regarding the most important prognostic indicators. 5. There was blinding of all subjects. 6. There was blinding of all therapists/researchers who administered the therapy/protocol. 7. There was blinding of all assessors who measured at least one key outcome. 8. Measures of at least one key outcome were obtained from more than 85% of the subjects initially allocated to groups. 9. All subjects for whom outcome measures were available received the treatment or control condition as allocated or, where this was not the case, data for at least one key outcome were analyzed by “intention to treat”. 10. The results of between-group statistical comparisons were reported for at least one key outcome. 11. The study provided both point measures and measures of variability for at least one key outcome.

Study	1 ^a^	2	3	4	5	6	7	8	9	10	11	Total
Boguszewski et al. [26]	1	1	0	1	0	0	0	1	1	1	1	6
Hodgson et al. [27]	1	1	0	1	0	0	0	1	1	1	1	6
Junker and Stöggl [29]	1	1	0	1	0	0	0	1	1	1	1	6
Kasahara et al. [30]	1	1	0	1	0	0	1	1	1	1	1	7
Le Gal et al. [28]	1	0	0	1	0	0	0	1	1	1	1	5
Ozden et al. [31]	1	1	0	1	0	0	0	1	1	1	1	6
Shalamzari et al. [32]	1	1	0	1	0	0	0	1	1	1	1	6
Stovern et al. [33]	1	1	0	1	0	1	1	1	1	1	1	8

## Data Availability

Not applicable.

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
