# Peer review of "The Effects of Foam Rolling Training on Performance Parameters: A Systematic Review and Meta-Analysis including Controlled and Randomized Controlled Trials"

_ijerph, 2022, doi:10.3390/ijerph191811638_

Round 1

Reviewer 1 Report

The comment can be found in attached pdf file.

Reviewer 2 Report

Authors are advised to consult the PRISMA 2020 Declaration, in its updated version for systematic reviews, to at least mention it in the text and check whether there have been any substantial changes. It is also recommended that the authors specify in the text the number of manuscripts obtained after discarding those that do not meet their inclusion and exclusion criteria. Systematic reviews usually give more details in this regard in the written part. However, the flow chart is correct and synthesizes well the results obtained in the search.

These are minor suggestions because in general it is a complete article that clearly describes the objective of the study, its methodology and the results obtained. 

Reviewer 3 Report

Authors must find additional literature in Chochrane, EMbase.

Authors should use the new PRISMA Flow Diagram.

Authors must obtain a registered number from an official site such as PROSPERO and describe it in the manisript.

Author should present the results of applying ROB2.

Round 2

Reviewer 3 Report

1. You must include P (people or person) in your PICO.

Foam Rolling Training Effects on Performance Parameters. A 2 Systematic Review and Meta-Analysis including controlled- 3 and randomized controlled trials

-> Effects of Foam Rolling Training on Performance Parameters (in Healthy Person) : A Systematic Review and Meta-Analysis of randomized controlled trials 

2. The conclusion should be described in the abstract.

3. Risk of Bias should be presented.

ex)

https://www.researchgate.net/figure/Risk-of-bias-of-each-study-included-in-this-meta-analysis-A-Risk-of-bias-summary_fig2_350928444